# Equivalence is All: A Unified View for Self-supervised Graph Learning

**Yejiang Wang** [1 2]  **Yuhai Zhao** [1 2]  **Zhengkui Wang** [3]  **Ling Li** [4]  **Jiapu Wang** [5]  **Fangting Li** [1 2]  **Miaomiao Huang** [1 2]  **Shirui Pan** [6]  **Xingwei Wang** [1]

## Abstract

Node equivalence is common in graphs, such as computing networks, encompassing automorphic equivalence (preserving adjacency under node permutations) and attribute equivalence (nodes with identical attributes). Despite their importance for learning node representations, these equivalences are largely ignored by existing graph models. To bridge this gap, we propose a GrAph self-supervised Learning framework with Equivalence (GALE) and analyze its connections to existing techniques. Specifically, we: 1) unify automorphic and attribute equivalence into a single equivalence class; 2) enforce the equivalence principle to make representations within the same class more similar while separating those across classes; 3) introduce approximate equivalence classes with linear time complexity to address the NP-hardness of exact automorphism detection and handle node-feature variation; 4) analyze existing graph encoders, noting limitations in message passing neural networks and graph transformers regarding equivalence constraints; 5) show that graph contrastive learning are a degenerate form of equivalence constraint; and 6) demonstrate that GALE achieves superior performance over baselines.

## 1. Introduction

Node equivalence (Lorrain & White, 1971; Bafai, 1977) is common in real-world graphs such as computing networks, where resources form nodes, and their dependencies form

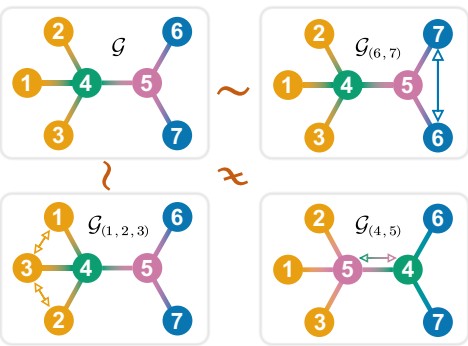

*Figure 1.* An example of automorphic equivalence. Hollow double arrows indicate node permutations in graph $\mathcal{G}$, shown in cycle notation (e.g., (1, 2, 3) means 2→1, 3→2, 1→3).

edges. In terms of graph structure, automorphic equivalence represents a strict form of equivalence: there exists a global permutation that maps nodes within the same equivalence class to each other (Cameron & Mary, 2004). As shown in Figure 1, we perform permutations on the nodes of the original graph $\mathcal{G}$. Here, $\mathcal{G}_{(i,j,...,k)}$ represents a permuted graph under the cycle notation of permutation, For instance, $\mathcal{G}_{(1,2,3)}$ means nodes 1, 2, and 3 are cyclically permuted (node 2 is mapped to node 1, node 3 to node 2, and node 1 to node 3). We can observe that $\mathcal{G} = \mathcal{G}_{(6,7)} = \mathcal{G}_{(1,2,3)}$, indicating that these nodes are structurally equivalent with respect to the graph topology. In contrast, nodes 4 and 5 are not equivalent, as swapping them would not preserve their adjacency relations.

Automorphic equivalence (AE) is a fundamental concept in various scientific domains such as chemistry and network analysis (Faulon, 1998; Friedkin & Johnsen, 1997). For instance, AE is a key indicator of similarity in social status and behavior in social networks (Everett, 1985). Furthermore, for graphs with node attributes, another type of equivalence can be defined based on attribute similarity. Specifically, a new equivalence class can be constructed by measuring the distance between attribute vectors: nodes with a close distance are grouped into the same equivalence class.

Naturally, we would expect graph representation models to preserve and reflect these node equivalences in their encoded node representations. However, we find that few studies ex-

[1]School of Computer Science and Engineering, Northeastern University, China [2]Key Laboratory of Intelligent Computing in Medical Image of Ministry of Education, Northeastern University, China [3]InfoComm Technology Cluster, Singapore Institute of Technology, SIT X NVIDIA AI Centre, Singapore [4]Shanxi University, China [5]Hefei University of Technology, China [6]Griffith University, Australia. Correspondence to: Yuhai Zhao <zhaoyuhai@mail.neu.edu.cn>.

plicitly consider these equivalences in graph representation learning, particularly in the widely studied paradigm of self-supervised graph learning, such as graph contrastive learning (Thakoor et al., 2022; Liu et al., 2024a; Wan et al., 2024). This method typically augments the original graph (e.g., by randomly removing edges) to generate two contrasting views. Contrastive constraints are then applied to encourage the representations of the same node (positive views) in both augmented graphs to be as similar as possible, while ensuring that representations of different nodes (negative views) become dissimilar. However, this paradigm overlooks the equivalence relations in terms of graph structure and node attributes. Existing methods focus solely on aligning the representations of the same node across augmented views, pushing apart those of different nodes, while often neglecting the fact that nodes similar to a given node are not limited to that node alone. Consequently, certain similarities between different nodes are frequently ignored.

To address these issues, we propose a novel equivalence-based self-supervised graph representation learning framework (GALE) grounded in the equivalence principle: node representations within the same equivalence class should be more similar, while those in different equivalence classes should be as dissimilar as possible. Specifically, 1) we define two equivalence relations—automorphic equivalence based on graph structure and attribute equivalence based on node attributes—and combine them into a unified equivalence class; 2) Then we enforce the equivalence principle, ensuring the node representations within the same equivalence class are similar, while those in different classes are dissimilar; 3) We introduce approximate equivalence classes with linear time complexity to address the NP-hard nature of exact automorphism detection and handle practical challenges such as attribute noise or slight variations that still indicate similarity, which make strict equivalence impractical in real-world graphs; 4) We note that message passing neural networks (MPNNs) make equivalent nodes similar but risk over-similarity for non-equivalent nodes, while most position encoding methods in graph transformers fail to incorporate the automorphic equivalence constraint; 5) From the equivalence perspective, we show that the current graph contrastive learning paradigm is a degenerate form of equivalence constraint, where each equivalence class is limited to a single node; 6) We demonstrate that GALE surpasses SOTA algorithms through experiments on benchmark datasets.

## 2. Related Work

**Equivalence in Graphs.** The concept of node equivalence traces its roots to structural sociology and algebraic graph theory (Wasserman & Faust, 1994). In social network analysis, Lorrain and White's *structural equivalence* (Lorrain & White, 1971; Fortunato, 2010) formalized the idea that

nodes sharing identical connectivity patterns occupy equivalent roles, forming the basis for role discovery in networks. However, its strict definition limits its applicability in complex networks. A broader concept of equivalence, *automorphic equivalence* (AE) (Lauri & Scapellato, 2016), emerged from graph automorphism theory, where two nodes are equivalent if there exists a permutation of the graph's vertices such that adjacency relations are preserved (Bafai, 1977). AE has proven fundamental across scientific domains, e.g., in social networks, AE identifies individuals with identical social statuses or influence patterns (Everett, 1985; Friedkin & Johnsen, 1997). In chemistry, symmetric atoms in molecules (e.g., carbon atoms in benzene rings) are automorphically equivalent, explaining their identical chemical properties (Faulon, 1998).

**Self-supervised Graph Learning.** Graph learning has attracted significant attention in recent years (Koh et al., 2024; Wang et al., 2024a;b; Liu et al., 2024b; Wang et al., 2024c). Graph self-supervised learning has emerged as a dominant paradigm for learning representations from unlabeled graph data (Wang et al., 2023; Zhao et al., 2025). Early approaches focused on preserving structural proximities through matrix factorization (Belkin & Niyogi, 2001), random walk objectives (Grover & Leskovec, 2016), or deep autoencoders (Cao et al., 2016). The recent surge in graph contrastive learning (GCL) methods (Cai et al., 2023; Li et al., 2022; Thakoor et al., 2022) has centered around maximizing agreement between positive views of the same node while repelling negative samples. Contrastive loss encourages similar representations for the same node across augmented views, while separating different nodes. However, it treats each node as only equivalent to itself, overlooking structural or attribute-based equivalence.

## 3. Background

**Notations.** Let $G = (V, E)$ be a graph with $n$ nodes and $m$ edges, where $V$ denotes the set of nodes and $E$ represents the set of edges. If node attributes are available, we denote the node feature matrix as $\boldsymbol{X} \in \mathbb{R}^{n \times d}$, where $d$ represents the dimension of the node attributes.

**Problem Formulation.** In this work, we focus on learning the representation $\mathbf{z}_v \in \mathbb{R}^q$ for each node $v$ in an *unsupervised* manner, where $q$ denotes the output dimension.

**Equivalence Relation and Equivalence Class.** Let $S$ be a non-empty set (in this paper, the set of interest is the node set $V$), and let $R$ be a relation on $S$. We say that $R$ is an *equivalence relation* on $S$ if and only if $R$ satisfies the following properties: 1) *Reflexivity*: Every element is related to itself under $R$ (i.e., $x \simeq_R x$ for all $x \in S$). 2) *Symmetry*: If $x \in S$ is related to $y \in S$ under $R$, then $y$ is related to $x$ under $R$ (i.e., $x \simeq_R y \Rightarrow y \simeq_R x$). 3)

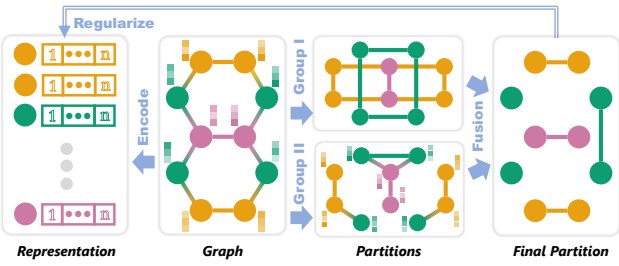

*Figure 2.* Overview of GALE, combining an encoder and equivalence constraints from structural and attribute-based partitions. The constraints regularize the encoder's node representations.

*Transitivity*: If $x \in S$ is related to $y \in S$ and $y$ is related to $z \in S$ under $R$, then $x$ is related to $z$ under $R$ (i.e., $x \simeq_R y \land y \simeq_R z \Rightarrow x \simeq_R z$). For example, the identity relation $(=)$ on a set $S$ is an equivalence relation. For each $x \in S$, the *equivalence class* of $x$ determined by $R$, denoted as $[x]$, is defined as $[x] = \{y \in S : x \simeq_R y\}$.

**Partition and Induced Partition.** Let $\mathcal{P}$ be a family of subsets of $S$. We say that $\mathcal{P}$ is a *partition* of $S$ if and only if the following conditions are satisfied: 1) If $X \in \mathcal{P}$, then $X \neq \varnothing$ (i.e., no subset in the partition is empty); 2) If $X \in \mathcal{P}$ and $Y \in \mathcal{P}$, then either $X = Y$ or $X \cap Y = \varnothing$ (i.e., subsets in the partition are pairwise disjoint); 3) $\bigcup_{X \in \mathcal{P}} X = S$ (i.e., the union of all subsets in the partition covers the entire set $S$). It can be proven that the equivalence classes determined by any equivalence relation on a set form a partition of that set. This partition is referred to as the *induced partition* from the equivalence relation.

**Automorphism and Orbit.** A permutation $\pi$ on a set $V$ rearranges its elements as $V_\pi = \{v_\pi\}$. We use cycle notation to represent permutations. All permutations of $V$ form a symmetric group $S_n$. A *graph automorphism* of a graph $G$ is a permutation $\pi \in S_n$ that preserves the edge relations of $G$, i.e., $E = E_\pi \triangleq \{(u_\pi, v_\pi) \mid (u, v) \in E\}$. Every graph has a trivial automorphism, the so-called *identity automorphism*, which maps each vertex to itself. The set of all automorphisms of a graph forms a group, called the *automorphism group* of the graph, denoted as $\mathrm{Aut}(G)$. The elements (permutations) in the automorphism group of a graph define an equivalence relation on the node set $V$:

$$u \simeq_{auto} v \iff \exists \pi \in \mathrm{Aut}(G), \ u_\pi = v \qquad (1)$$

This equivalence relation, also known as a *symmetry relation*, partitions the node set $V$ into non-overlapping equivalence classes called *orbits*.

## 4. Learning with Equivalence Classes

In this Section, we introduce a novel equivalence-based self-supervised graph learning framework (GALE), as shown in Figure 2. GALE consists of a dual-branch structure. One

branch involves a *graph encoder*, which encodes the graph to obtain node representations; we use a graph neural network (GNN) for this purpose. The other branch focuses on inducing equivalence-class *partitions* of the node set. Based on the *equivalence principle*—nodes within the same equivalence class are considered similar, while nodes in different classes are considered dissimilar—we impose regularization constraints on the node representations and optimize the encoder with an equivalence loss. In our work, we construct two types of equivalence-class partitions: 1) *Group I*: Calculated based on the graph's adjacency information, capturing the graph's structural automorphism. 2) *Group II*: Calculated based on the attribute information of nodes, capturing feature equality. We fuse these partitions into a unified node constraint network, ensuring the learned representations capture both structural and attribute-based equivalences.

### 4.1. Automorphic Equivalence from Structure

A graph typically encompasses two components: the relationships between nodes (edges) and the properties associated with each node. We establish equivalence relations and their corresponding partitions on both of these aspects. First, we focus on adjacency. As discussed earlier, automorphic equivalence represents a strict node equivalence relation based on the graph's structure. Indeed, real-world graph data often exhibits a substantial number of nodes with automorphic equivalence.

To verify this, we present statistics on the proportion of nodes/graphs with automorphic equivalence in 3 benchmark network datasets (Kipf & Welling, 2017) and 3 graph datasets (Morris et al., 2020). For network data (single graph), we calculate the ratio of nodes in non-singleton orbits (the number of nodes is greater than one) to the total number of nodes. For graph datasets (multiple graphs), we compute the ratio of graphs with non-trivial automorphisms. The results, shown in Table 1, demonstrate that a significant proportion of nodes/graphs exhibit non-trivial automorphic equivalence, highlighting its prevalence in real-world datasets.

*Table 1.* Proportions of automorphically equivalent nodes (left three) and graphs (right four) in network and graph-level datasets.

| Data | Cora | Citeseer | PubMed | MUTAG | DD | IMDB-B | COLLAB |
|------|------|----------|--------|-------|------|--------|--------|
| Ratio | 18.4% | 46.2% | 45.7% | 100% | 96.3% | 100% | 99.9% |

To compute the exact automorphisms of a graph, we adopt classical algorithms such as *bliss* or *nauty* (Junttila & Kaski, 2011; McKay & Piperno, 2014). For example, *bliss* is an efficient method for computing the automorphism group of real-world graphs by leveraging symmetry-breaking and backtracking search. Once the automorphism orbits are obtained, we construct the automorphic equivalence partition $\mathcal{P}_{\mathrm{auto}} = \{C_i\}$, where $C_i$ represents an orbit.

### 4.2. Attribute Equivalence from Nodes

To construct a partition of the graph based on node attributes, we define an equivalence relation among nodes such that nodes with identical attributes are grouped into the same equivalence class:

$$u \simeq_{attr} v \iff \mathbf{x}_u = \mathbf{x}_v, \quad \forall u, v \in V \tag{2}$$

which ensures that the partition reflects the inherent equation of nodes in terms of their attributes. This equivalence relation induces a partition $\mathcal{P}_{attr} = \{D_i\}$, where each subset $D_i \subseteq V$ contains all nodes with the same attribute values.

### 4.3. Fusion of Equivalences

With the two partitions established, we aim to create a refined node partition by fusion. Generally, both the structural role and the attributes of a node are equally important for its characterization. For example, consider a citation network (where nodes represent papers and edges represent citations): two papers might be cited by similar papers (graph structure), yet their topics (attributes) could be different. Therefore, we seek a refined partition that preserves the equivalence relationships from both perspectives. To accomplish this, we construct the fused partition by intersecting the structure-based and attribute-based partition:

$$\mathcal{P}_{\text{fuse}} = \{C_i \cap D_j \mid C_i \in \mathcal{P}_{\text{auto}}, D_j \in \mathcal{P}_{\text{attr}}, C_i \cap D_j \neq \emptyset\} \tag{3}$$

It inherits characteristics from both partitions, ensuring consistency with both graph structure and node attributes. Each fused equivalence class $F = C_i \cap D_j$ contains nodes that are equivalent under both graph automorphisms and node attribute equivalence. The resulting partition satisfies: 1) $\bigcup_{F \in \mathcal{P}_{\text{fuse}}} F = V$; 2) $F_k \cap F_l = \emptyset$ for $k \neq l$. We now prove that the relation $\simeq_{\text{fuse}}$, which induces the fused partition $\mathcal{P}_{\text{fusion}}$, satisfies the properties of an equivalence relation.

**Theorem 4.1.** *The fusion relation defined as $u \simeq_{fuse} v$ is an equivalence relation.*

*Proof.* To show that $\simeq_{\text{fuse}}$ is an equivalence relation, we verify the three properties: 1) Reflexivity: For any node $u \in V$, we have $u \simeq_{\text{auto}} u$ (since automorphic equivalence is reflexive) and $u \simeq_{\text{attr}} u$ (since attribute equivalence is reflexive), Therefore, $u \simeq_{\text{fuse}} u$, satisfying reflexivity. 2) Symmetry: If $u \simeq_{\text{fuse}} v$, then $u \simeq_{\text{auto}} v$ and $u \simeq_{\text{attr}} v$. By the symmetry of $\simeq_{\text{auto}}$ and $\simeq_{\text{attr}}$, we have $v \simeq_{\text{auto}} u$ and $v \simeq_{\text{attr}} u$, proving symmetry. 3) Transitivity: If $u \simeq_{\text{fuse}} v$ and $v \simeq_{\text{fuse}} w$, then $u \simeq_{\text{auto}} v$, $v \simeq_{\text{auto}} w$, and $u \simeq_{\text{attr}} v$, $v \simeq_{\text{attr}} w$. By the transitivity of $\simeq_{\text{auto}}$ and $\simeq_{\text{attr}}$, we have $u \simeq_{\text{auto}} w$ and $u \simeq_{\text{attr}} w$. Hence, transitivity is established. $\square$

### 4.4. Equivalence Loss

To encourage nodes within the same fused equivalence class to have similar representations and nodes from different equivalence classes to have dissimilar representations, we define a loss function based on the similarity between node embeddings.

**Intra-class Loss.** Let $\mathcal{P}_{\text{fuse}} = \{F_i\}$ represent the fused equivalence partition, and let $\mathbf{z}_u$ denote the embedding of node $u$. The intra-class loss is defined to encourage the embeddings of nodes within the same equivalence class $F_i$ to be similar:

$$\mathcal{L}_{\text{intra}} = \sum_i \sum_{u,v \in F_i} -\mathcal{D}(\mathbf{z}_u, \mathbf{z}_v) \tag{4}$$

where $\mathcal{D} : \mathbb{R}^q \times \mathbb{R}^q \to \mathbb{R}$ is a discriminator that maps two views to an agreement score. In this work, we use the standard inner product $\mathcal{D}(\mathbf{z}_i, \mathbf{z}_j) = \mathbf{z}_i^T \mathbf{z}_j$.

**Inter-class Loss.** The inter-class loss is designed to ensure that embeddings of nodes from different equivalence classes are dissimilar. For nodes $s \in F_i$ and $t \in F_j$ with $i \neq j$, we penalize high similarity between their embeddings. The inter-class loss is defined as:

$$\mathcal{L}_{\text{inter}} = \sum_{i \neq j} \sum_{s \in F_i, t \in F_j} \mathcal{D}(\mathbf{z}_s, \mathbf{z}_t) \tag{5}$$

Finally, the equivalence loss is defined as:

$$\mathcal{L}_{\text{equiv}} = \mathcal{L}_{\text{intra}} + \mathcal{L}_{\text{inter}} \tag{6}$$

## 5. Approximate Equivalence Classes

In real-world graphs, strict equivalence relationships, such as automorphic or exact attribute-based equivalence, are often impractical due to the computational cost of determining exact automorphisms and the rarity of complete attribute equality caused by noise or data variations. To overcome the challenges, we introduce approximate equivalence classes, which relax the strict requirements of exact equivalence.

### 5.1. Approximate Automorphic Equivalence

Determining the exact automorphism group of a graph is computationally intractable, as it is the NP-hard problem. To overcome this, we propose an efficient approximation using PageRank vectors, which leverage their inherent relationship to approximate automorphic equivalence effectively.

The PageRank algorithm (Langville & Meyer, 2004) assigns a score to each node in a graph, measuring its relative importance based on the graph structure. Let $\mathbf{P} \in \mathbb{R}^{n \times n}$ denote the Markov transition matrix of the graph, where $P_{ij} = 1/\deg(i)$ if $(i, j) \in E$, and $P_{ij} = 0$ otherwise, with $\deg(i)$ representing the degree of node $i$. The PageRank vector $\mathbf{r} \in \mathbb{R}^n$ is the stationary distribution of a random walk with teleportation, based on the transition probabilities $\mathbf{P}$, computed as:

$$\mathbf{r} = \alpha \mathbf{P}^\top \mathbf{r} + (1 - \alpha)\mathbf{v} \tag{7}$$

where $\alpha \in (0,1)$ is the teleportation probability, $\mathbf{v} \in \mathbb{R}^n$ is a personalization vector (typically uniform). The PageRank vector assigns a score $r_i$ to each node $i \in V$, reflecting its importance based on local connectivity and global graph structure. The following lemma establishes the relationship between automorphisms and PageRank.

**Lemma 5.1** (Ghorbani et al. (2021)). *If two vertices $u, v \in V$ are automorphically equivalent in $G$, then $u$ and $v$ have the same PageRank score:*

$$u \simeq_{auto} v \implies r_u = r_v \tag{8}$$

We leverage this property to approximate automorphic equivalence by grouping nodes with similar PageRank scores. While nodes with equal PageRank scores are not necessarily automorphically equivalent, they often exhibit structural similarity in terms of their importance within the graph, providing an efficient approximation. To verify this, we conduct experiments on 8 benchmark datasets: Cora, Citeseer, Pubmed (Kipf & Welling, 2017), Wiki-CS, Amazon-Computers, Amazon-Photo, Coauthor-CS, and Coauthor-Physics (Shchur et al., 2018). Empirical results show the alignment between the true and approximate equivalence partitions, measured by the Variation of Information (Meilă, 2007) (with values closer to 0 indicating better alignment and 0 indicating perfect alignment). The detailed results are presented in Table 2, where we find that for all the data, the approximate partitions closely match the true ones, validating the effectiveness of our approximation. Here, we use $\alpha = 0.85$; for an analysis of different parameter settings, please refer to the Appendix.

*Table 2.* Variation of Information (VI) for alignment between equivalences $\simeq_{auto}$ and $\simeq_{PR}$ on eight benchmark data. Lower VI values (0 indicating perfect alignment) are shaded in darker green.

| Data | Cora | CiteSeer | PubMed | WikiCS | Amz-Comp. | Amz-Photo | Co-CS | Co-Phy. |
|------|------|----------|--------|--------|-----------|-----------|-------|---------|
| VI($\downarrow$) | 0.021 | 0.047 | 0 | 0.001 | 0.003 | 0.004 | 0 | 0.001 |

Computing PageRank has a time complexity of $\mathcal{O}(m)$ for graphs, where $m$ denotes the number of edges, which is significantly more efficient than exact automorphism computation (Kondaveeti et al., 2024).

### 5.2. Approximate Attribute Equivalence

In real-world networks, nodes often exhibit attributes that are not perfectly identical but are instead only slightly different. Strictly enforcing exact equality of node attributes when defining equivalence classes can lead to overly fragmented partitions, as even minor differences in attributes result in nodes being classified into different equivalence classes. To address this issue, we propose relaxing the equivalence constraint by using a similarity threshold based on the Euclidean distance between node attribute vectors. This allows

nodes with sufficiently similar attributes to be grouped into the same equivalence class. For nodes $u, v \in V$, we define the approximate attribute equivalence relation $\simeq_{attr}^{\epsilon}$ as:

$$u \simeq_{attr}^{\epsilon} v \iff \|\mathbf{x}_u - \mathbf{x}_v\|_2 \leq \epsilon \tag{9}$$

where $\epsilon \geq 0$ is a small positive threshold that controls the degree of similarity required for nodes to be considered equivalent. The relaxed partition allows nodes with similar attributes (within the threshold $\epsilon$) to be grouped together, even if their attributes are not exactly identical. Clearly, when $\epsilon = 0$, it reduces to strict equivalence. When $\epsilon > 0$, it is not a strict equivalence, though the resulting node sets may be called similarity groups.

### 5.3. Complexity Analysis

The time complexity of the GALE involves four components: 1) Automorphic equivalence can be approximated using PageRank, which runs in $\mathcal{O}(m)$; 2) Attribute-based equivalence requires grouping nodes by attributes, with a complexity of $\mathcal{O}(n^2)$; 3) Fusion of equivalences involves intersecting equivalence classes, scaling as $\mathcal{O}(n)$ in practice due to the typically small number and size of partitions. 4) Equivalence loss computation with a worst-case complexity of $\mathcal{O}(n^2)$. Overall, the dominant term is $\mathcal{O}(n^2 + m)$.

## 6. Equivalence and Graph Encoders

In this section, we discuss how mainstream graph encoders, message passing neural networks (MPNNs) and graph transformers, interact with the concept of equivalence.

### 6.1. MPNNs and Automorphic Equivalence

MPNNs (Wu et al., 2020) and their variants, such as Graph Convolutional Networks (GCNs), rely on the principle of neighborhood aggregation to learn node representations. The neighborhood aggregation mechanism of MPNNs inherently aligns with the goal of making automorphically equivalent nodes have similar representations. For example, consider two automorphically equivalent nodes $u$ and $v$ and their adjacency structures are identical. During the message-passing process, $u$ and $v$ will aggregate similar information from their neighbors. This property ensures that MPNNs can naturally preserve the similarity of node representations within the same equivalence class.

However, MPNNs cannot guarantee that dissimilarity between different equivalence classes remains valid. As the number of layers increases, the representations of all nodes in the graph tend to converge, leading to the over-smoothing problem. This phenomenon reduces the discriminative power of the learned node embeddings. Common over-smoothing mitigations, like residual connections or dynamic neighborhoods, typically overlook node equivalence.

To address this, explicit constraints based on equivalence classes can be incorporated into MPNN models. These constraints enforce inter-class dissimilarity while preserving intra-class similarity, helping to alleviate the over-smoothing problem. To demonstrate this, Figure 3 shows the classification results of GCN and GALE with increasing depth on the Cora and CiteSeer dataset. GCN's performance declines sharply with more layers, but GALE with equivalence constraints significantly mitigates the over-smoothing issue.

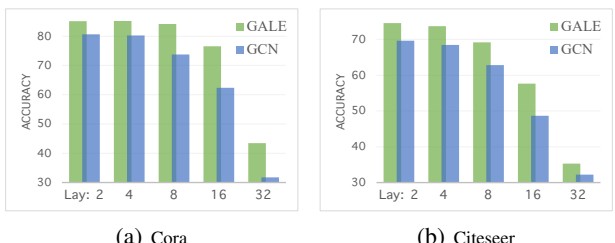

(a) Cora                    (b) Citeseer

*Figure 3.* Accuracy of GCN and GALE with increasing depth.

### 6.2. Graph Transformers and Equivalence

Graph transformers (Min et al., 2022) take a fundamentally different approach to learning node representations compared to MPNNs. Instead of relying on neighborhood aggregation based on adjacency structure, graph transformers use node positional embeddings to inject structural information and infer relationships between nodes through their positions and attributes. The node representations are then updated using a fully-connected self-attention mechanism.

While graph transformers can capture global relationships, they do not inherently ensure that automorphically equivalent nodes are represented similarly. The effectiveness of capturing automorphic equivalence depends on the quality of the positional embeddings, which may not always align with the graph's symmetries. Moreover, the global nature of the attention mechanism, which lacks local constraints, means that nodes within the same equivalence class might not remain sufficiently similar, and there's no inherent mechanism to ensure dissimilarity between nodes in different equivalence classes. As a result, while graph transformers excel at encoding global structures, they may struggle to consistently respect automorphic equivalence without additional constraints or interventions.

**Positional Encoding and AE.** Therefore, node positional embeddings, which are solely based on the graph's structure, are essential for reflecting the equivalence relationships within that structure. To evaluate whether common positional encoding (PE) methods align with automorphic equivalence, we propose a validation procedure that compares the ground truth automorphic equivalence with those derived from positional embeddings, using a distance threshold $\epsilon_{\text{pe}} = 10^{-4}$ to construct equivalence classes. Specifically,

we conduct experiments on 8 benchmark graph datasets using 5 PE methods: Laplace positional encoding (LapPE) (Kreuzer et al., 2021), Random walk positional encoding (RWSE) (Dwivedi et al., 2022), SignNet (Lim et al., 2023), ElstaticSE, and HKdiagSE (Rampášek et al., 2022). We compute the Variation of Information (VI) (Meilă, 2007), and the results are shown in Table 3. Our findings indicate that most existing PE methods do not truly adhere to automorphic equivalence, and in some cases, deviate significantly. This may be because these methods focus primarily on ensuring that neighboring nodes are assigned similar positions (locally), while neglecting the global equivalences within the graph.

*Table 3.* Variation of Information between automorphic and positional encoding-based equivalence (lower is better, 0 is perfect).

| Methods | Cora | CiteSeer | PubMed | WikiCS | Amz-Comp. | Amz-Photo | Co-CS | Co-Phy |
|---|---|---|---|---|---|---|---|---|
| LapPE | 7.309 | 6.201 | 0.063 | 9.101 | 9.368 | 8.808 | 0.147 | 1.804 |
| RWSE | 0.073 | 0.345 | 0.138 | 0.238 | 0.158 | 0.050 | 0.035 | 0.023 |
| SignNet | 7.547 | 6.966 | 0.669 | 4.879 | 5.068 | 4.508 | 0.963 | 0.932 |
| HKdiagSE | 0.007 | 0.037 | 0.002 | 3.446 | 3.590 | 2.025 | 0.048 | 1.308 |
| ElstaticSE | 0.345 | 1.533 | 0.096 | 0.764 | 0.483 | 0.182 | 0.004 | - |

## 7. Equivalence in Graph Contrastive

In this section, we analyze how equivalence classes play a role in graph contrastive learning (GCL).

### 7.1. Analysis of Graph Contrastive Learning

GCL starts by creating two views of the original graph $G_1$ through small perturbations, such as edge deletion, resulting in an augmented graph $G_2$. These two views, $G_1$ and $G_2$, are assumed to have minimal differences and preserve label invariance. For each node $i$, the goal of the contrastive loss is to make the node's representations across the two views $\mathbf{z}_i^1$ (from $G_1$) and $\mathbf{z}_i^2$ (from $G_2$) more similar (positive pairs), while ensuring that the representations of different nodes $\mathbf{z}_j^1$ (from $G_1$, $j \neq i$) and $\mathbf{z}_i^2$ (from $G_2$) are as dissimilar as possible (negative pairs). The contrastive loss can be formulated as follows:

$$\mathcal{L}_{\text{con}} = -\frac{1}{n} \sum_{i=1}^{n} \log \frac{\exp(\text{sim}(\mathbf{z}_i^1, \mathbf{z}_i^2)/\tau)}{\sum_{j=1}^{n} \exp(\text{sim}(\mathbf{z}_i^1, \mathbf{z}_j^2)/\tau)} \quad (10)$$

where $\text{sim}(\cdot, \cdot)$ is a similarity function, $\tau$ is a temperature.

From the perspective of equivalence classes, current GCL methods implicitly assume a trivial equivalence class structure, where each node is treated as its own equivalence class. Specifically, the contrastive loss enforces that each node's representation remains consistent across views (within the same trivial equivalence class) while being distinct from the representations of all other nodes (across different classes).

Clearly, the assumption of trivial equivalence classes limits the ability of GCL to capture richer graph structures. Treat-

ing each node as a separate equivalence class ignores similarities between nodes, such as automorphic equivalence or attribute-based similarity.

## 7.2. Role of Augmentation and Contrastive

GCL primarily relies on two mechanisms: graph augmentation and a contrastive loss that implicitly enforces equivalence class constraints. To disentangle their respective contributions, we conduct ablation studies on the loss in Eq. (10) using five benchmark datasets (Kipf & Welling, 2017). In the first ablation (*w/o aug*), we remove graph augmentations and perform contrastive learning on two identical views of the graph (i.e., $G_1 = G_2$), using a single shared encoder (instead of two). This design isolates the effect of augmentation by ensuring there is no variation in the contrastive loss other than the inherent randomness in training. In the second ablation (*w/o Pos*), we modify the contrastive loss by removing positive pairs—i.e., the term encouraging the similarity of $z_i^1$ and $z_i^2$—and retain only the negative sample term. This effectively disrupts the equivalence class constraint, as nodes are no longer required to maintain consistent representations across views. Table 4 presents the corresponding results.

*Table 4.* Node classification accuracy (%) without augmentation.

| Dataset | Cora | Citeseer | PubMed | Amz-Comp. | Coa-CS. |
|---|---|---|---|---|---|
| Aug+Pos | 83.26 ± 0.56 | 73.20 ± 0.37 | 81.33 ± 0.58 | 88.85 ± 0.30 | 92.71 ± 0.33 |
| w/o Aug | 82.12 ± 0.60 | 72.79 ± 0.21 | 80.86 ± 0.72 | 88.42 ± 0.44 | 92.22 ± 0.29 |
| w/o Pos | 76.10 ± 0.39 | 64.47 ± 0.48 | 80.37 ± 0.51 | 86.27 ± 0.21 | 89.38 ± 0.36 |

The results in the table show that removing graph augmentations has minor impact on performance, indicating that augmentations are not the primary driver of GCL's effectiveness. In contrast, removing positive pairs significantly reduces performance across most data, highlighting the importance of equivalence constraints in the contrastive loss.

## 8. Comparison with the SOTA Methods

### 8.1. Experimental Setup

**Datasets.** We evaluate the proposed model on both node classification and graph classification tasks. For node classification, we use 8 benchmark datasets: Cora, Citeseer, Pubmed (Kipf & Welling, 2017), Wiki-CS, Amazon-Computers, Amazon-Photo, Coauthor-CS, and Coauthor-Physics (Shchur et al., 2018). For graph classification, we evaluate on 8 datasets from the TUDataset benchmark (Morris et al., 2020), including NCI1, PROTEINS, DD, MUTAG, COLLAB, RDT-B, RDT-M5K, and IMDB-B.

**Baselines.** We compare our method against a wide range of baseline methods across both node-level and graph-level tasks. For graph-level task: 1) two *supervised learning methods*, including GCN (Kipf & Welling, 2017) and GIN (Xu et al., 2018); 2) four *kernel-based methods*, including SP (Borgwardt & Kriegel, 2005), GK (Shervashidze et al., 2009), WL (Shervashidze et al., 2011), DGK (Yanardag & Vishwanathan, 2015); 3) three *unsupervised methods*, including NODE2VEC (Grover & Leskovec, 2016), SUB2VEC (Adhikari et al., 2018), GRAPH2VEC (Narayanan et al., 2017); 4) five *self-supervised graph contrastive learning*, including INFOGRAPH (Sun et al., 2020), GRAPHCL (You et al., 2020), AD-GCL (Suresh et al., 2021), JOAOV2 (You et al., 2021), RGCL (Li et al., 2022), SIMGRACE (Xia et al., 2022), SEGA (Wu et al., 2023), and AUTOGCL (Yin et al., 2022); For node-level tasks: 1) *Supervised learning methods*, including MLP and GCN (Kipf & Welling, 2017); 2) *Graph embedding methods*, including DEEPWALK (Perozzi et al., 2014) and NODE2VEC (Grover & Leskovec, 2016); 3) *Graph contrastive learning methods*, including VGAE (Kipf & Welling, 2016), DGI (Velickovic et al., 2019), GMI (Peng et al., 2020), MVGRL (Hassani & Khasahmadi, 2020), GRACE (Zhu et al., 2020), GCA (Zhu et al., 2021), BGRL (Thakoor et al., 2022), CCA-SSG (Zhang et al., 2021) and SUGRL(Mo et al., 2022).

**Protocol.** We follow the standard evaluation protocol of previous state-of-the-art self-supervised learning methods. For node classification, we report the mean accuracy on the test set after 50 runs of training. Pretrained node embeddings are used to train a linear neural network for classification. The dataset is split into 10%/10%/80% for training, validation, and testing, respectively. For graph classification, we evaluate the learned graph representations using a linear SVM classifier. We report the mean 10-fold cross-validation accuracy across 5 runs. For each training fold, the linear SVM is tuned using cross-validation, and the best mean accuracy is reported. The dataset is split into 80%/10%/10% for training, validation, and testing, respectively.

**Implementation Details.** We implement both GALE and its variant GALE-APR using PyTorch Geometric. The key difference is that GALE uses Nauty (McKay & Piperno, 2014) for exact automorphisms, while GALE-APR employs PageRank equivalence with $\alpha = 0.85$. For node attributes, both rely on $\simeq_{\text{attr}}^{\epsilon}$ with $\epsilon \in \{0, 10^{-7}, \ldots, 10^{-1}\}$. We adopt the Adam optimizer, tuning learning rates $\{0.0001, 0.001, 0.01\}$, batch sizes $\{16, 64, 128, 256, 512\}$. As iterations increase, Intra-class loss (4) dominates, causing the total loss (6) to become imbalanced and too small. We can add Softplus after the discriminator to mitigate this.

### 8.2. Results and Analysis

**Performance on Graph-level Tasks.** We evaluate whether GALE and its variant GALE-APR can outperform state-of-the-art methods on multiple graph-level benchmarks. Table 5 summarizes the results for both supervised and unsupervised baselines. Overall, GALE achieves top-tier perfor-

*Table 5.* Supervised and unsupervised classification accuracy (%) on TU datasets. The highest unsupervised performance is shaded in *green*, and the second-highest performance is shaded in *light green*. '–' means that the results are unavailable.

| Model | NCI1 | PROTEINS | DD | MUTAG | COLLAB | RDT-B | RDT-M5K | IMDB-B |
|---|---|---|---|---|---|---|---|---|
| GCN | $80.20 \pm 0.14$ | $74.92 \pm 0.33$ | $76.24 \pm 0.14$ | $85.63 \pm 0.24$ | $79.01 \pm 0.18$ | $50.00 \pm 0.10$ | $20.00 \pm 0.10$ | $70.45 \pm 0.37$ |
| GIN | $82.75 \pm 0.19$ | $76.28 \pm 0.28$ | $78.91 \pm 0.13$ | $89.47 \pm 0.16$ | $80.23 \pm 0.19$ | $92.44 \pm 0.25$ | $56.50 \pm 0.24$ | $73.70 \pm 0.60$ |
| SP | $73.53 \pm 0.16$ | $75.07 \pm 0.54$ | $>1d$ | $85.25 \pm 0.24$ | – | $64.13 \pm 0.04$ | $39.64 \pm 0.02$ | $55.62 \pm 0.02$ |
| GK | $66.06 \pm 0.12$ | $71.67 \pm 0.55$ | $78.53 \pm 0.03$ | $81.71 \pm 0.21$ | $71.81 \pm 0.31$ | $77.32 \pm 0.02$ | $41.06 \pm 0.08$ | $65.93 \pm 0.10$ |
| WL | $80.01 \pm 0.50$ | $72.92 \pm 0.56$ | $79.78 \pm 0.36$ | $80.76 \pm 0.30$ | $69.30 \pm 0.42$ | $68.82 \pm 0.41$ | $46.06 \pm 0.21$ | $72.30 \pm 0.44$ |
| DGK | $79.98 \pm 0.36$ | $73.21 \pm 0.61$ | $74.79 \pm 0.32$ | $87.51 \pm 0.65$ | $64.43 \pm 0.48$ | $78.35 \pm 0.43$ | $41.49 \pm 0.32$ | $67.09 \pm 0.37$ |
| NODE2VEC | $54.93 \pm 0.16$ | $57.58 \pm 0.36$ | – | $72.62 \pm 1.02$ | $56.12 \pm 0.02$ | – | – | $50.25 \pm 0.09$ |
| SUB2VEC | $52.82 \pm 0.15$ | $53.06 \pm 0.56$ | $54.33 \pm 0.24$ | $61.17 \pm 1.59$ | $55.26 \pm 0.15$ | $71.48 \pm 0.42$ | $36.68 \pm 0.42$ | $55.34 \pm 0.15$ |
| GRAPH2VEC | $73.21 \pm 0.18$ | $73.33 \pm 0.21$ | $79.32 \pm 0.29$ | $83.28 \pm 0.93$ | $71.10 \pm 0.54$ | $75.78 \pm 0.96$ | $46.86 \pm 0.26$ | $71.16 \pm 0.05$ |
| INFOGRAPH | $76.33 \pm 0.79$ | $74.27 \pm 0.43$ | $73.02 \pm 1.31$ | $88.65 \pm 1.09$ | $70.41 \pm 0.34$ | $83.06 \pm 0.82$ | $53.45 \pm 1.10$ | $72.97 \pm 0.49$ |
| GRAPHCL | $77.54 \pm 0.34$ | $74.41 \pm 0.35$ | $78.49 \pm 0.36$ | $86.62 \pm 1.23$ | $71.28 \pm 1.16$ | $89.50 \pm 0.73$ | $55.85 \pm 0.33$ | $71.29 \pm 0.38$ |
| AD-GCL | $74.15 \pm 0.62$ | $73.33 \pm 0.36$ | $75.74 \pm 0.68$ | $88.70 \pm 1.66$ | $72.01 \pm 0.53$ | $90.04 \pm 0.76$ | $54.55 \pm 0.40$ | $70.35 \pm 0.59$ |
| JOAOV2 | $78.36 \pm 0.21$ | $74.02 \pm 0.53$ | $77.36 \pm 0.29$ | $87.23 \pm 1.12$ | $69.05 \pm 0.22$ | $86.36 \pm 0.30$ | $55.77 \pm 0.23$ | $70.12 \pm 0.25$ |
| RGCL | $78.14 \pm 0.89$ | $75.11 \pm 0.58$ | $78.77 \pm 0.52$ | $87.62 \pm 1.16$ | $70.85 \pm 0.63$ | $90.27 \pm 0.54$ | $56.39 \pm 0.37$ | $71.67 \pm 0.76$ |
| SIMGRACE | $79.06 \pm 0.15$ | $75.26 \pm 0.24$ | $77.13 \pm 0.88$ | $89.23 \pm 1.25$ | $71.26 \pm 0.51$ | $89.41 \pm 0.33$ | $55.91 \pm 0.23$ | $71.22 \pm 0.30$ |
| SEGA | $79.15 \pm 0.52$ | $76.00 \pm 0.68$ | $78.81 \pm 0.33$ | $90.23 \pm 0.89$ | $74.10 \pm 0.62$ | $90.12 \pm 0.46$ | $56.08 \pm 0.40$ | $73.60 \pm 0.33$ |
| AUTOGCL | $78.39 \pm 0.58$ | $69.80 \pm 0.38$ | $75.76 \pm 0.93$ | $85.22 \pm 1.43$ | $71.40 \pm 0.26$ | $86.56 \pm 1.31$ | $55.68 \pm 0.23$ | $72.26 \pm 0.34$ |
| GALE | $80.04 \pm 0.43$ | $81.15 \pm 0.31$ | $80.63 \pm 0.78$ | $91.12 \pm 0.93$ | $75.95 \pm 0.21$ | $90.10 \pm 0.49$ | $56.56 \pm 0.40$ | $77.66 \pm 0.11$ |
| GALE-APR | $79.07 \pm 0.30$ | $81.11 \pm 0.55$ | $80.04 \pm 0.48$ | $90.52 \pm 1.20$ | $75.11 \pm 0.29$ | $90.03 \pm 0.78$ | $55.89 \pm 0.29$ | $76.78 \pm 0.27$ |

*Table 6.* Mean node classification accuracy for supervised and unsupervised models. The highest unsupervised performance is shaded in *green*, and the second-highest performance is shaded in *light green*. OOM indicates Out-Of-Memory on a 24GB GPU.

| Model | Cora | CiteSeer | PubMed | WikiCS | Amz-Comp. | Amz-Photo | Coauthor-CS | Coauthor-Phy. |
|---|---|---|---|---|---|---|---|---|
| MLP | $47.92 \pm 0.41$ | $49.31 \pm 0.26$ | $69.14 \pm 0.34$ | $71.98 \pm 0.42$ | $73.81 \pm 0.21$ | $78.53 \pm 0.32$ | $90.37 \pm 0.19$ | $93.58 \pm 0.41$ |
| GCN | $81.54 \pm 0.68$ | $70.73 \pm 0.65$ | $79.16 \pm 0.25$ | $93.02 \pm 0.11$ | $86.51 \pm 0.54$ | $92.42 \pm 0.22$ | $93.03 \pm 0.31$ | $95.65 \pm 0.16$ |
| DEEPWALK | $70.72 \pm 0.63$ | $51.39 \pm 0.41$ | $73.27 \pm 0.86$ | $74.42 \pm 0.13$ | $85.68 \pm 0.07$ | $89.40 \pm 0.11$ | $84.61 \pm 0.22$ | $91.77 \pm 0.15$ |
| NODE2VEC | $71.08 \pm 0.91$ | $47.34 \pm 0.84$ | $66.23 \pm 0.95$ | $71.76 \pm 0.14$ | $84.41 \pm 0.14$ | $89.68 \pm 0.19$ | $85.16 \pm 0.04$ | $91.23 \pm 0.07$ |
| VGAE | $77.27 \pm 0.86$ | $67.46 \pm 0.20$ | $76.02 \pm 0.52$ | $75.55 \pm 0.22$ | $86.40 \pm 0.30$ | $92.13 \pm 0.12$ | $92.10 \pm 0.31$ | $94.43 \pm 0.20$ |
| DGI | $82.24 \pm 0.63$ | $71.82 \pm 0.61$ | $76.80 \pm 0.30$ | $75.42 \pm 0.17$ | $84.05 \pm 0.42$ | $91.62 \pm 0.37$ | $92.14 \pm 0.55$ | $94.54 \pm 0.52$ |
| GMI | $82.40 \pm 0.57$ | $71.74 \pm 0.12$ | $79.28 \pm 0.94$ | $74.79 \pm 0.16$ | $82.24 \pm 0.39$ | $90.81 \pm 0.15$ | OOM | OOM |
| MVGRL | $83.37 \pm 0.65$ | $73.29 \pm 0.36$ | $80.33 \pm 0.61$ | $77.55 \pm 0.06$ | $87.45 \pm 0.17$ | $91.77 \pm 0.21$ | $92.24 \pm 0.31$ | $95.30 \pm 0.13$ |
| GRACE | $81.88 \pm 0.84$ | $71.13 \pm 0.42$ | $80.88 \pm 0.13$ | $79.37 \pm 0.24$ | $86.48 \pm 0.24$ | $92.20 \pm 0.16$ | $92.90 \pm 0.27$ | $95.25 \pm 0.26$ |
| GCA | $82.41 \pm 0.55$ | $71.56 \pm 0.19$ | $80.73 \pm 0.23$ | $78.26 \pm 0.39$ | $87.92 \pm 0.33$ | $92.35 \pm 0.53$ | $92.65 \pm 0.32$ | $95.52 \pm 0.21$ |
| BGRL | $81.86 \pm 0.32$ | $72.10 \pm 0.31$ | $80.65 \pm 0.42$ | $79.28 \pm 0.45$ | $89.21 \pm 0.47$ | $92.28 \pm 0.44$ | $92.73 \pm 0.41$ | $95.31 \pm 0.26$ |
| SUGRL | $83.29 \pm 0.38$ | $73.11 \pm 0.22$ | $81.96 \pm 0.49$ | $78.88 \pm 0.35$ | $88.98 \pm 0.20$ | $92.87 \pm 0.19$ | $92.84 \pm 0.24$ | $94.80 \pm 0.24$ |
| CCA-SSG | $84.17 \pm 0.44$ | $73.27 \pm 0.30$ | $81.91 \pm 0.41$ | $77.67 \pm 0.29$ | $88.88 \pm 0.22$ | $93.14 \pm 0.43$ | $93.23 \pm 0.16$ | $95.29 \pm 0.11$ |
| GALE | $85.36 \pm 0.15$ | $74.58 \pm 0.18$ | $85.06 \pm 0.27$ | $82.15 \pm 0.22$ | $90.60 \pm 0.19$ | $94.51 \pm 0.34$ | $93.46 \pm 0.28$ | $95.91 \pm 0.38$ |
| GALE-APR | $85.14 \pm 0.26$ | $75.35 \pm 0.10$ | $84.57 \pm 0.11$ | $82.07 \pm 0.36$ | $90.27 \pm 0.20$ | $94.38 \pm 0.46$ | $93.45 \pm 0.32$ | $95.86 \pm 0.23$ |

mance: for instance, on PROTEINS it outperforms the best self-supervised baseline SEGA by 5.15%, and on IMDB-B it surpasses GIN by 3.96%. Meanwhile, GALE-APR exhibits a similar pattern, slightly underperforming GALE on a few datasets but still outperforming most baselines. In summary, both GALE and GALE-APR demonstrate superior performance across a wide range of graph-level tasks.

**Performance under Node-level.** Table 6 summarizes our results on eight node-level benchmarks, showing that both GALE and GALE-APR not only outperform the strongest unsupervised models but also exceed the supervised baseline GCN across most datasets. For instance, on PubMed, GALE achieves 85.06%, surpassing the best unsupervised method SUGRL by over 3.1% and supervised method GCN by 5.9%. Meanwhile, GALE-APR stays highly competitive, for example on Coauthor-CS it closely matching GALE, thus confirming the effectiveness of our approach.

## 9. Conclusion

This paper proposes a unified self-supervised graph representation framework, GALE, which integrates structural automorphism and attribute equivalence to construct joint equivalence classes. Such combined equivalences are frequently encountered in real-world scenarios, including computing networks. To address computational bottlenecks and noise interference, we design an efficient approximation strategy with linear complexity. GALE preserves the similarity of equivalent nodes while enforcing separation between representations of non-equivalent nodes. Analysis reveals the inherent limitations of mainstream graph models, including over-smoothing in MPNNs, failure to respect automorphism constraints in Transformer positional encodings, and the degeneration of graph contrastive learning into single-node equivalence. Experimental results demonstrate the superiority of GALE.

## Acknowledgments and Disclosure of Funding

This project was in part supported by the following projects: the National Natural Science Foundation of China (No.62432003, No.92267206).

## Impact Statement

This paper aims to advance Graph Learning, with potential societal impacts that don't require specific highlighting here.

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

## A. Impact of $\alpha$ on PageRank Equivalence

To analyze the impact of different $\alpha$ values in PageRank on the resulting equivalence classes, we conduct experiments on four benchmark datasets: Cora, Citeseer, Amz-Photo, and Coauthor-CS. The $\alpha$ values range from 0.1 to 0.9 with an increment of 0.1, and for each dataset, we compute the Variation of Information (VI) (Meilă, 2007) between equivalence partitions obtained under different $\alpha$ settings. This results in $9 \times 9$ heatmaps for each data, which are visualized in Figure 4.

Our findings reveal that the differences between equivalence classes under varying $\alpha$ values are minimal across all datasets. For instance, in the Cora dataset, the VI values are consistently in the range of $0.001$, indicating negligible variation. Similarly, in the Photo dataset, we observe 33 perfect matches (VI = 0) across different parameter pairs, further demonstrating the robustness of PageRank equivalence classes to changes in $\alpha$.

These results suggest that the choice of $\alpha$ has little effect on the overall structure of the equivalence classes derived from PageRank scores. This stability highlights the reliability of using PageRank for approximating equivalence relations in graphs, regardless of the specific damping factor chosen.

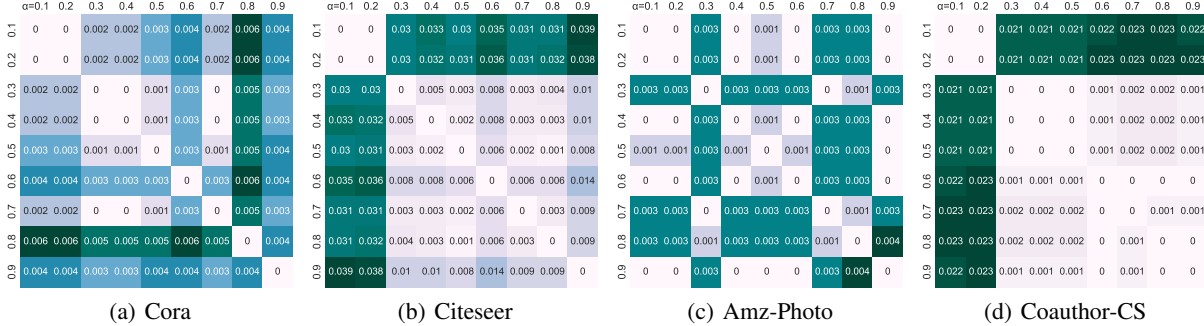

|     | (a) Cora | (b) Citeseer | (c) Amz-Photo | (d) Coauthor-CS |

*Figure 4.* Heatmaps showing the Variation of Information (VI) values between PageRank equivalence classes obtained under different $\alpha$ values, for four datasets. Lighter colors indicate lower VI values, signifying greater similarity between equivalence partitions.

## B. Related Work[Extended]

**Equivalence in Graphs Neural Networks.** There have been recent advancements in incorporating automorphic equivalence (AE) (Kondor & Trivedi, 2018) into graph neural networks (GNNs). GRAPE (Xu et al., 2021) introduces ego-centered automorphic equivalence to differentiate nodes within localized neighborhoods using subgraph template, while Autobahn (Thiede et al., 2021) performs local equivariant convolutions over subgraph automorphism groups, improving adaptability for tasks like molecular property prediction. Pearce-Crump & Knottenbelt (2024) primarily focuses on the theoretical analysis and generalization of neural network design by characterizing learnable, linear, Aut(G)-equivariant neural layers, leveraging bilabelled graphs to compute representations. In contrast, our work introduces a self-supervised learning framework that not only considers structural equivalence but also incorporates node attribute information, enabling the joint learning of structural roles and attribute semantics to offer a comprehensive perspective.

## C. Ablation Study

### C.1. Contribution of Automorphic and Feature Equivalence

To evaluate the importance of incorporating both automorphic and node attributes equivalence in our framework, we conduct ablation experiments on eight benchmark datasets: Cora, Citeseer, Pubmed, Wikics, Computers, Photo, CS, and Physics. We compare the performance of the full model (*origin*) with two variants: 1) **w/o auto**: This variant removes automorphic equivalence information, relying solely on node attributes for equivalence class partitioning. 2) **w/o feat**: This variant removes node attribute equivalence, relying only on automorphic equivalence for equivalence class partitioning.

The results, summarized in Table 7, show that excluding either automorphic or node attributes equivalence leads to performance degradation across all datasets. The experimental results reveal the following key observations. 1) **Effect of removing automorphic equivalence (*w/o auto*)**: When automorphic equivalence is excluded, the performance consistently declines across all datasets. This is because solely relying on node attributes ignores the structural information that helps

distinguish nodes with similar attributes but different structural roles in the graph. For example, in a social network, two users may both have similar attributes, such as age, profession, or location, but one might be a central influencer (hub node) while the other is a peripheral user with few connections. Ignoring automorphic equivalence would fail to capture the distinct structural importance of these users. 2) **Effect of removing feature equivalence(*w/o feat*)**: Excluding attributes equivalence also leads to a performance drop, sometimes more significant than *w/o auto*, particularly in datasets like Citeseer. Without feature equivalence, nodes with very different characteristics may be grouped together solely based on automorphic equivalence, reducing the model's discriminative power. For instance, in a social network, nodes with high structural similarity (e.g., both being part of tightly connected communities) but vastly different attributes (e.g., different professions) could be inappropriately treated as equivalent.

These findings underscore the importance of integrating both automorphic equivalence and node attribute information. The combination of structural and attribute-based perspectives ensures a more comprehensive representation of node equivalence, effectively capturing both the roles and features of nodes in the graph.

*Table 7.* Ablation study results on 8 datasets. *origin* represents the full model, while *w/o auto* and *w/o feat* represent the ablated models.

| Methods | Cora | CiteSeer | PubMed | WikiCS | Amz-Comp. | Amz-Photo | Co-CS | Co-Phy |
|---|---|---|---|---|---|---|---|---|
| *origin* | $85.36 \pm 0.15$ | $74.58 \pm 0.18$ | $85.06 \pm 0.27$ | $82.15 \pm 0.22$ | $90.60 \pm 0.19$ | $94.51 \pm 0.34$ | $93.46 \pm 0.28$ | $95.91 \pm 0.38$ |
| *w/o auto* | $82.25 \pm 0.09$ | $72.44 \pm 0.12$ | $83.63 \pm 0.30$ | $81.03 \pm 0.34$ | $88.30 \pm 0.28$ | $93.07 \pm 0.36$ | $92.37 \pm 0.41$ | $94.46 \pm 0.13$ |
| *w/o feat* | $80.42 \pm 0.24$ | $64.72 \pm 0.30$ | $83.22 \pm 0.21$ | $80.87 \pm 0.27$ | $88.84 \pm 0.16$ | $94.50 \pm 0.29$ | $91.98 \pm 0.17$ | $93.19 \pm 0.44$ |

## C.2. Evaluation of Equivalence Fusion Operator

This section presents an ablation study to empirically evaluate and justify our choice of using the intersection operator for fusing node equivalences in our proposed model (referred to as *origin*). An alternative fusion strategy, based on the union operator, was implemented and compared. The primary goal is to assess how these different fusion methods impact overall model performance, particularly in capturing meaningful node similarities derived from diverse equivalence criteria (e.g., those based on automorphic properties and node attributes).

Table 8 summarizes the performance results on the eight benchmark datasets when employing either the intersection-based (*origin*) or union-based (*union*) fusion approach. The experimental results presented in Table 8 demonstrate that the proposed model, which employs intersection fusion, consistently outperforms the *union* fusion approach. This is largely because emphasizing one equivalence type (either automorphic equivalence or node attributes) by using the union tends to neglect the contributions from the other. In contrast, the intersection method ensures that both types of equivalences jointly contribute, leading to a more robust and comprehensive representation of node similarity. These findings justify our choice of the intersection method for fusing equivalences.

*Table 8.* Ablation study results comparing intersection (*origin* model) and union (*union* model) for fusing equivalences on 8 datasets.

| Methods | Cora | CiteSeer | PubMed | WikiCS | Amz-Comp. | Amz-Photo | Co-CS | Co-Phy |
|---|---|---|---|---|---|---|---|---|
| *origin* | $85.36 \pm 0.15$ | $74.58 \pm 0.18$ | $85.06 \pm 0.27$ | $82.15 \pm 0.22$ | $90.60 \pm 0.19$ | $94.51 \pm 0.34$ | $93.46 \pm 0.28$ | $95.91 \pm 0.38$ |
| *union* | $77.21 \pm 0.22$ | $64.37 \pm 0.12$ | $83.22 \pm 0.19$ | $80.96 \pm 0.34$ | $87.86 \pm 0.25$ | $93.59 \pm 0.28$ | $91.55 \pm 0.15$ | $94.93 \pm 0.36$ |

## D. Rand Index Results

In the main body of the paper, we evaluated the alignment between true automorphic equivalence classes ($\simeq_{auto}$) and our PageRank-based approximation ($\simeq_{PR}$) using the Variation of Information (VI) metric (detailed in Table 2). To provide a more comprehensive validation of this alignment, in this section we presents supplementary results using the Rand Index (RI) (Rand, 1971). The RI is another widely recognized metric for comparing the similarity between two data partitions.

The Rand Index assesses agreement by considering all possible pairs of elements and checking if each pair is consistently placed (i.e., either together in both partitions or separated in both partitions). It produces a score ranging from 0 to 1, where 1 signifies perfect agreement between the two partitions, and values closer to 1 indicate a high degree of similarity. Table 9 reports the RI scores for the alignment between $\simeq_{auto}$ and $\simeq_{PR}$ across the same eight benchmark datasets analysed in the main text. The consistently high RI values, approaching 1 for nearly all datasets, strongly corroborate the findings obtained with the VI metric.

*Table 9.* Rand Index (RI) for alignment between equivalences $\simeq_{\text{auto}}$ and $\simeq_{\text{PR}}$ on eight benchmark datasets (higher is better, 1 is perfect).

| Data | Cora | CiteSeer | PubMed | WikiCS | Amz-Comp. | Amz-Photo | Co-CS | Co-Phy |
|---|---|---|---|---|---|---|---|---|
| **RI** ($\uparrow$) | 0.99948 | 0.99687 | 0.99999 | 0.99999 | 0.99999 | 0.99998 | 0.99999 | 0.99999 |

## E. Sensitivity Analysis of PageRank Teleportation

To evaluate the impact of hyperparameter choices on model performance, we conducted a sensitivity analysis on the Cora dataset with respect to the PageRank teleportation parameter ($\alpha$). The parameter $\alpha$ was varied from 0.1 to 0.9, and the corresponding model performance is reported in Table 10.

*Table 10.* Sensitivity analysis of the PageRank teleportation parameter ($\alpha$) on the Cora dataset.

| Dataset | 0.1 | 0.2 | 0.3 | 0.4 | 0.5 | 0.6 | 0.7 | 0.8 | 0.9 |
|---|---|---|---|---|---|---|---|---|---|
| Cora | $85.33 \pm 0.19$ | $85.35 \pm 0.15$ | $85.27 \pm 0.20$ | $85.38 \pm 0.12$ | $85.23 \pm 0.19$ | $85.36 \pm 0.17$ | $85.30 \pm 0.09$ | $85.35 \pm 0.15$ | $85.29 \pm 0.24$ |

As shown in Table 10, the model's accuracy remains stable across a wide range of $\alpha$ values, indicating that the performance is not sensitive to the choice of this hyperparameter.

## F. Limitation and Futrue Work

While our core contribution focuses on static homogeneous graphs, we acknowledge two related limitations. First, GALE currently assumes a static graph setting, meaning that for dynamic graphs with evolving structures or features, the equivalence classes may change over time, necessitating incremental updates to partitions—a challenge common to equivalence-based methods. Second, our present work targets homogeneous graphs; extending our equivalence definitions to heterogeneous graphs (e.g., those with multi-typed nodes/edges) would require additional design considerations, such as type-specific automorphism constraints. We consider these aspects as important directions for our future work.

