# OpenReview forum: "Equivalence is All: A Unified View for Self-supervised Graph Learning"
_ICML.cc/2025/Conference — ICML 2025 oral_

### Official Review · Reviewer_Smmv · 2025-03-10

**Overall Recommendation:** 5

**Summary:**

This paper proposes a novel self-supervised graph learning framework grounded in the principle of node equivalence, which unifies structural (automorphic) and attribute-based equivalence classes to learn robust node representations. The work is well-motivated, offering a principled unification of structural and attribute-based node relationships, backed by theoretical insights, scalable approximations, and comprehensive empirical validation.

**Claims And Evidence:**

The claims are well-motivated and largely supported by theoretical and empirical evidence.

**Essential References Not Discussed:**

The paper makes valuable contributions but omits some reference, such as [1] formalizes equivalence (e.g., group-equivariant networks) in convolution neural networks. Citing these would strengthen the paper’s positioning.
Refs:
[1] Kondor, R., & Trivedi, S. On the generalization of equivariance and convolution in neural networks to the action of compact groups. ICML 2018.

**Experimental Designs Or Analyses:**

The experimental designs and results are largely sound.

**Methods And Evaluation Criteria:**

The proposed methods and evaluation criteria are appropriate and well-aligned with the paper’s goals. The core innovations, unified equivalence classes, scalable approximations, are tested on standard benchmarks.

**Other Comments Or Suggestions:**

The main text inconsistently uses "automorphism equivalence" and "automorphic equivalence" (e.g., in the third paragraph of the introduction). It is recommended to standardize the terminology throughout the paper.
Additionally, citing classic references such as [1].

**Other Strengths And Weaknesses:**

Strength:

S1: The paper is well-written and easy to understand. I appreciate the ambition behind unifying structural and attribute equivalence. It bridges graph-theoretic principles with modern representation learning, offering a fresh lens to rethink graph SSL.

S2: The method’s design is well-founded—approximating automorphic equivalence via PageRank and relaxing attribute constraints strike a pragmatic balance between rigor and scalability.

S3: The experiment is well-conducted and validates its performance.

Weakness:
W1: The impact of hyperparameter choices (e.g., PageRank teleportation parameter) on model performance is not sufficiently discussed. A sensitivity analysis of these parameters would strengthen the empirical section.

W2: Incomplete figure and table explanations. For example, the cycle notation in Figure 1 is not explained in the caption, making it difficult for non-expert readers to understand. In the tables, the highlighted colors (e.g., shades of green) lack a legend, leaving their meaning unclear.

**Questions For Authors:**

1. The magnitude difference between Intra-class Loss and Inter-class Loss may cause optimization bias (e.g., Intra-class Loss dominates). Would it be better to introduce a temperature coefficient or weight adjustment to balance them?

**Relation To Broader Scientific Literature:**

The paper sits at the intersection of graph theory and SSL, advancing these areas by providing a unified equivalence framework that subsumes existing SSL methods; introducing scalable approximations to bridge classical graph isomorphism concepts with modern deep learning; and offering actionable insights to improve GNNs and transformers via symmetry preservation.

This work aligns with the broader trend toward principled graph SSL (e.g., invariance theories in GCL) but stands out by rigorously formalizing equivalence as a first-class citizen in representation learning.

**Theoretical Claims:**

I reviewed the theoretical claims, including proofs in the main paper.

---

> ### Author Rebuttal · Authors · 2025-04-01
>
> **Q1. The paper makes valuable contributions but omits some reference, such as [1] formalizes equivalence (e.g., group-equivariant networks) in convolution neural networks. Citing these would strengthen the paper’s positioning. Refs: [1] Kondor, R., & Trivedi, S. On the generalization of equivariance and convolution in neural networks to the action of compact groups. ICML 2018.**
>
> >R1: Thank you for the valuable suggestion. We will include this reference in the revised manuscript to strengthen our positioning.
>
> **Q2. The impact of hyperparameter choices (e.g., PageRank teleportation parameter) on model performance is not sufficiently discussed. A sensitivity analysis of these parameters would strengthen the empirical section.**
>
> >R2: Thank you for your insightful comment. As requested, we have performed a sensitivity analysis on the Cora dataset with respect to the PageRank teleportation parameter ($\alpha$). We varied $\alpha$ from 0.1 to 0.9 and recorded the performance of our model. The results are shown in the table below:
>
> *Table I: Sensitivity analysis of PageRank teleportation parameter (α) on the Cora dataset*
>
> | $\alpha$ |     0.1      |     0.2      |     0.3      |     0.4      |     0.5      |     0.6      |     0.7      |     0.8      |     0.9      |
> | :------: | :----------: | :----------: | :----------: | :----------: | :----------: | :----------: | :----------: | :----------: | :----------: |
> |   Cora   | 85.33 ± 0.19 | 85.35 ± 0.15 | 85.27 ± 0.20 | 85.38 ± 0.12 | 85.23 ± 0.19 | 85.36 ± 0.17 | 85.30 ± 0.09 | 85.35 ± 0.15 | 85.29 ± 0.24 |
>
> >From the table, it is evident that our algorithm’s performance is not sensitive to the choice of $\alpha$; the accuracy remains almost consistent across all values. We will include these details in the updated empirical section of the manuscript.
>
> **Q3. Incomplete figure and table explanations. For example, the cycle notation in Figure 1 is not explained in the caption, making it difficult for non-expert readers to understand. In the tables, the highlighted colors (e.g., shades of green) lack a legend, leaving their meaning unclear.**
>
> >R3: Thank you for your suggestion. We will revise the caption for Figure 1 to provide a clearer explanation. The updated caption is as follows:
>
> >_Figure 1. An example of an automorphic equivalence. The hollow double arrows indicate the permutation of nodes in graph $\mathcal{G}$. The permutation is expressed in cycle notation (e.g., (1 2 3) indicates that node 1 maps to node 2, node 2 to node 3, and node 3 back to node 1)._
>
> >In addition, the highlighted colors use lighter shades to represent smaller numbers. A detailed explanation will be provided in the revised manuscript.
>
> **Q4. The main text inconsistently uses "automorphism equivalence" and "automorphic equivalence" (e.g., in the third paragraph of the introduction). It is recommended to standardize the terminology throughout the paper. Additionally, citing classic references such as [1].**
>
> >R4: Thank you for your valuable suggestion. We will standardize the terminology throughout the paper by consistently using "automorphic equivalence." Additionally, we will include reference [1] in the revised manuscript.
>
> **Q5. The magnitude difference between Intra-class Loss and Inter-class Loss may cause optimization bias (e.g., Intra-class Loss dominates). Would it be better to introduce a temperature coefficient or weight adjustment to balance them?**
>
> >R5: Thank you for the insightful comment. Rather than introducing additional hyperparameters like a temperature coefficient or weight adjustments, we address the imbalance by applying a Softplus activation after the discriminator. This approach is also described in the Implementation Details section of the manuscript. Once again, thank you for your thoughtful feedback.

---

> > ### Comment · Reviewer_Smmv · 2025-04-03
> >
> > Thank you for the author's response, resolving my concerns. This is an interesting and excellent paper.

---

> > > ### Author Response · Authors · 2025-04-03
> > >
> > > Thank you for your positive feedback and valuable review. Your input has strengthened our work.

---

### Official Review · Reviewer_jbhi · 2025-03-10

**Overall Recommendation:** 3

**Summary:**

This paper introduces a self-supervised graph learning framework that unifies automorphic equivalence (structural symmetry) and attribute equivalence (node feature similarity) into a cohesive representation learning paradigm. The work bridges structural and feature-based node similarities, offering a principled approach to self-supervised graph learning while addressing scalability and practicality challenges.

Thanks to the author for the reply, this solved most of my problems, I will keep my score.

**Claims And Evidence:**

Yes, I believe that the majority of the claims are well-supported by evidence.

**Essential References Not Discussed:**

N/A.

**Experimental Designs Or Analyses:**

I have reviewed most of the experimental design and analysis, which appear comprehensive and reasonable. For the equivalence class matching evaluation, I recommend adding other classic metrics, such as the Rand Index.

**Methods And Evaluation Criteria:**

Yes, the proposed methods are well-suited to the problem.

**Other Comments Or Suggestions:**

Terms like "automorphic equivalence" and "automorphism equivalence" are used interchangeably, risking confusion.

**Other Strengths And Weaknesses:**

Strengths

1. Originality:
   - The unification of automorphic equivalence and attribute equivalence into a single framework is novel and addresses a gap in graph representation learning.
   - The critique of graph contrastive learning as a "degenerate" equivalence constraint offers a fresh perspective.

2. Significance:
   - The work bridges graph theory and modern deep learning, providing a principled approach to self-supervised learning that aligns with real-world graph properties (e.g., social roles, molecular symmetries).
   - Experiments demonstrate consistent performance gains.

3. Clarity of Writing:
   - The paper is generally well-structured, with clear definitions of equivalence relations and a logical flow from motivation to experiments.
   - Figures and tables enhance readability.

Weaknesses

1. Writing and Presentation:
   - Inconsistent Terminology: Terms like "automorphic equivalence" and "automorphism equivalence" are used interchangeably, risking confusion.

2. Ablation:
   - The choice of intersection for fusing equivalences is not justified against alternatives (e.g., union). Ablation studies comparing fusion methods are missing.

3. Minor Issues:
   - The overview figure (Figure 2) lacks a clear explanation of how the encoder interacts with equivalence constraints.

**Questions For Authors:**

See in Other Strengths And Weaknesses.

**Relation To Broader Scientific Literature:**

The paper bridges graph theory, contrastive learning, and self-supervised paradiam design, offering a fresh perspective on equivalence-driven representation learning.

**Theoretical Claims:**

Yes, I have checked the correctness of the proofs for the theoretical claims.

---

> ### Author Rebuttal · Authors · 2025-04-01
>
> **Q1. I have reviewed most of the experimental design and analysis, which appear comprehensive and reasonable. For the equivalence class matching evaluation, I recommend adding other classic metrics, such as the Rand Index.**
>
> >R1: Thank you for the valuable feedback. We have added a table below that includes the Rand Index results for our algorithm. The Rand Index is a well-established metric for evaluating matching performance, where values closer to 1 indicate a perfect match. Our results show that across nearly all datasets, the Rand Index is almost equal to 1, which further validates the performance and effectiveness of our approximate method.
>
> *Table I: Rand Index (RI) for alignment between equivalences $\simeq_{\text{auto}}$ and $\simeq_{\text{PR}}$ on eight benchmark datasets (higher is better, 1 is perfect).*
>
> |  **Data**  | **Cora** | **CiteSeer** | **PubMed** | **WikiCS** | **Amz-Comp.** | **Amz-Photo** | **Co-CS** | **Co-Phy.** |
> | :--------: | :------: | :----------: | :--------: | :--------: | :-----------: | :-----------: | :-------: | :---------: |
> | **RI (↑)** | 0.99948  |   0.99687    |  0.99999   |  0.99999   |    0.99999    |    0.99998    |  0.99999  |   0.99999   |
>
> **Q2. Inconsistent Terminology: Terms like "automorphic equivalence" and "automorphism equivalence" are used interchangeably, risking confusion.**
>
> >R2: Thank you for highlighting the inconsistency in terminology. In the revised manuscript, we will standardize the terminology by uniformly using "automorphic equivalence" throughout the paper to ensure clarity and consistency.
>
> **Q3. The choice of intersection for fusing equivalences is not justified against alternatives (e.g., union). Ablation studies comparing fusion methods are missing.**
>
> >R3: Thank you for the insightful suggestion. We have added a table below that presents performance results on our datasets using the union method as an alternative for fusing equivalences. Our experiments indicate that the union approach performs markedly worse compared to the intersection method. This is largely because, as detailed in the ablation study in the appendix, emphasizing one equivalence type (either automorphic equivalence or node attributes) by using the union tends to neglect the contributions from the other. In contrast, the intersection method ensures that both types of equivalences jointly contribute, leading to a more robust and comprehensive representation of node similarity. These findings justify our choice of the intersection method for fusing equivalences.
>
> *Table II. Ablation study results on 8 datasets. Origin represents the full model, while union represent the ablated models.*
>
> | Model  |     Cora     |   Citeseer   |    Pubmed    |    Wikics    |  Computers   |    Photo     |      CS      |    Physics    |
> | :----: | :----------: | :----------: | :----------: | :----------: | :----------: | :----------: | :----------: | :-----------: |
> | origin | 85.36 ± 0.15 | 74.58 ± 0.18 | 85.06 ± 0.27 | 82.15 ± 0.22 | 90.60 ± 0.19 | 94.51 ± 0.34 | 93.46 ± 0.28 | 95.91 ± 0.38  |
> | union  | 77.21 ± 0.22 | 64.37 ± 0.12 | 83.22 ± 0.19 | 80.96 ± 0.34 | 87.86 ± 0.25 | 93.59 ± 0.28 | 91.55 ± 0.15 | 94.93  ± 0.36 |
>
> **Q4. The overview figure (Figure 2) lacks a clear explanation of how the encoder interacts with equivalence constraints.**
>
> >R4: Thank you for your valuable comment. In our approach, the encoder is designed to output a representation for each node in the graph. We then apply equivalence constraints to these representations to ensure that nodes which are equivalent (according to our defined criteria) are embedded in a consistent manner. This process helps the model to better capture both structural and attribute similarities. We will update the caption of Figure 2 in the revised manuscript to include a clear explanation of how the encoder interacts with the equivalence constraints.

---

### Official Review · Reviewer_LJki · 2025-03-12

**Overall Recommendation:** 3

**Summary:**

This paper presents a novel framework for self-supervised graph representation learning that emphasizes the importance of node equivalence. The authors propose GALE, which unifies automorphic equivalence (based on graph structure) and attribute equivalence (based on node attributes) into a single equivalence class. The framework enforces the equivalence principle, ensuring that node representations within the same equivalence class are similar while those in different classes are dissimilar. Key contributions include the introduction of approximate equivalence classes with linear time complexity to address computational challenges, an analysis of existing graph encoders' limitations regarding equivalence constraints, and the demonstration that graph contrastive learning paradigms are a degenerate form of equivalence constraints. The paper also shows that GALE outperforms state-of-the-art baselines through extensive experiments on benchmark datasets. The authors provide a comprehensive analysis of the connections between equivalence classes and existing techniques, highlighting the potential of their approach to improve the performance and effectiveness of graph representation learning models.

**Claims And Evidence:**

The claims made in this submission are generally well-supported by clear and convincing evidence.

**Essential References Not Discussed:**

The paper mentions connections to other fields such as social network analysis. However, it does not cite recent work that could provide additional context. For example, the paper by Santo Fortunato titled "Community detection in graphs" provides a comprehensive review of community detection methods in graphs.

**Experimental Designs Or Analyses:**

The experimental designs and analyses in this paper are generally sound and well-executed.

**Methods And Evaluation Criteria:**

The proposed methods and evaluation criteria in this paper are appropriate for the problem of self-supervised graph representation learning.

**Other Comments Or Suggestions:**

In the introduction, when referring to automorphic equivalence, the paper states that “permuting all nodes within the same equivalence class preserves the graph’s edge relations” However, strict graph automorphism involves a global permutation (an isomorphic mapping of the entire graph), rather than only permuting nodes within an equivalence class. This should be corrected to: "There exists a global permutation that maps nodes within the same equivalence class to each other."

**Other Strengths And Weaknesses:**

Pros:

1.The paper presents an unified framework for incorporating node equivalence into self-supervised graph representation learning. This combination of automorphic and attribute equivalence offers a interesting perspective on improving graph representation learning.
2.The proposed framework addresses a gap in existing graph representation learning methods by explicitly considering node equivalence. This is a contribution to the field.
3.The paper is well-written and well-structured, making it easy to follow the authors' reasoning and methodology.
4.The experiment is capable of validating its conclusion.

Cons:

1.The paper only discusses the over-smoothing issue in MPNNs but does not mention existing mitigation techniques (e.g., residual connections). It would be helpful to discuss the relationship between equivalence constraints and these techniques.
2.Although the introduction of approximate equivalence classes aims to address the computational complexity, a more in-depth discussion of the trade-offs between accuracy and efficiency would be valuable.

**Questions For Authors:**

1.The paper only discusses the over-smoothing issue in MPNNs but does not mention existing mitigation techniques (e.g., residual connections). It would be helpful to discuss the relationship between equivalence constraints and these techniques.
2. Although the introduction of approximate equivalence classes aims to address the computational complexity, a more in-depth discussion of the trade-offs between accuracy and efficiency would be valuable.

**Relation To Broader Scientific Literature:**

The paper is related to prior work on graph representation learning, social network analysis and self-supervised learning.

**Theoretical Claims:**

The theoretical claims made in the paper are logically sound.

---

> ### Author Rebuttal · Authors · 2025-04-01
>
> **Q1. The paper mentions connections to other fields such as social network analysis. However, it does not cite recent work that could provide additional context. For example, the paper by Santo Fortunato titled "Community detection in graphs" provides a comprehensive review of community detection methods in graphs.**
>
> >R1: Thank you for the helpful suggestion. We will include this reference "Community detection in graphs" in the revised version.
>
> **Q2. The paper only discusses the over-smoothing issue in MPNNs but does not mention existing mitigation techniques (e.g., residual connections). It would be helpful to discuss the relationship between equivalence constraints and these techniques.**
>
> >R2: Thank you for the suggestion. While residual connections help with gradient propagation, they do not explicitly distinguish between equivalent and non-equivalent nodes. Similarly, some methods dynamically alter neighbors in deeper GCNs to alleviate over-smoothing, but they also lack an explicit mechanism to differentiate between nodes and often require careful tuning. In contrast, our GALE approach explicitly enforces this distinction, making representations semantically meaningful. We will add this discussion to clarify the relationship between equivalence constraints and existing techniques in the revised version.
>
> **Q3. Although the introduction of approximate equivalence classes aims to address the computational complexity, a more in-depth discussion of the trade-offs between accuracy and efficiency would be valuable.**
>
> >R3: Thank you for the feedback. Our experimental results (see Tables 5 and 6) show that the approximate method may incur a slight precision drop compared to the exact method. However, it consistently outperforms current state-of-the-art methods, and in some datasets, it even surpasses the exact algorithm. This demonstrates that the efficiency gains come with only minimal trade-offs in accuracy, underscoring the effectiveness of our approach.
>
> **Q4. In the introduction, when referring to automorphic equivalence, the paper states that “permuting all nodes within the same equivalence class preserves the graph’s edge relations” However, strict graph automorphism involves a global permutation (an isomorphic mapping of the entire graph), rather than only permuting nodes within an equivalence class. This should be corrected to: "There exists a global permutation that maps nodes within the same equivalence class to each other."**
>
> >R4: Thank you for the insightful comment. We appreciate your insight and will incorporate this change in the revised version of the paper. Thanks again.

---

### Official Review · Reviewer_nMZa · 2025-03-12

**Overall Recommendation:** 4

**Summary:**

The paper introduces GALE, a self-supervised graph learning framework that unifies automorphic and attribute equivalence into a single node equivalence concept, enforcing intra-class similarity and inter-class dissimilarity through a novel loss function. The paper claims that by explicitly modeling and enforcing node equivalence, the proposed framework provides a more principled approach to self-supervised graph learning that improves upon traditional contrastive methods.

**Claims And Evidence:**

The paper’s claims are generally well supported by a combination of theoretical exposition, algorithmic design, and extensive empirical results.

**Essential References Not Discussed:**

To my best knowledge, all the core references are cited correctly.

**Experimental Designs Or Analyses:**

I have reviewed the Node and Graph Classification Experiments, Ablation Studies, Evaluation of Equivalence Alignment and the Over-smoothing Analysis. The experimental designs and analyses are sound.

**Methods And Evaluation Criteria:**

Yes.

**Other Comments Or Suggestions:**

Some typos:
- Line 75: Change "noting" to "note".
- Line 252: Change "indicates" to "indicating".
- Line 308 (right column): Change "a" to "an".

**Other Strengths And Weaknesses:**

**Strengths.**

1. The paper offers a new approach by unifying automorphic and attribute equivalence into a single framework, which is a very interesting perspective in self-supervised graph learning.

2. The paper provides new insights from an equivalence-class perspective, such as over-smoothing in MPNNs and limitations in current positional encoding schemes in Graph Transformers.

3. The paper is well-written.

4. Extensive experiments demonstrate the advantages of the proposed approach.

**Weaknesses.**

1. Using PageRank to approximate automorphic equivalence is an efficient approach, but nodes with similar PageRank scores are not necessarily structurally symmetric. Could this have impact on the model's performance?

2. The experimental tables (e.g., color-coded cells) may not fully convey information through text descriptions alone. It is recommended to add textual explanations in the paper, such as clarifying the meaning of "VI" values and specifying threshold ranges.

3. The paper would benefit from a discussion on the potential limitations of the model.

**Questions For Authors:**

1. The definition of approximate attribute equivalence does not appear to satisfy transitivity (i.e.,  $u \simeq v$ and $v \simeq w$ do not necessarily imply $u \simeq w$). Does this mean it does not form a strict equivalence relation? If so, it is recommended to use "similarity groups" instead.

**Relation To Broader Scientific Literature:**

The paper interweaves classical graph theory with self-supervised learning techniques and deep neural architectures.

- The paper extends traditional self-supervised graph learning methods, particularly graph contrastive learning (e.g., GRACE) by challenging the standard paradigm that treats each node as an isolated equivalence class.
- The idea of automorphic equivalence has long been studied in graph theory, with foundational works focusing on graph isomorphism and symmetry detection (e.g., using algorithms like Nauty and Bliss).

**Theoretical Claims:**

I have checked the theoretical claim (Theorem 4.1) and the proof process appears to be correct.

---

> ### Author Rebuttal · Authors · 2025-04-01
>
> **Q1. Using PageRank to approximate automorphic equivalence is an efficient approach, but nodes with similar PageRank scores are not necessarily structurally symmetric. Could this have impact on the model's performance?**
>
> >R1: We acknowledge that similar PageRank scores do not strictly guarantee structural symmetry. However, our experiments (see Table 2 in the paper) demonstrate that the approximate equivalence obtained via PageRank aligns almost perfectly with exact automorphic equivalence—indicated by Variation of Information (VI) values that are nearly zero (with 0 representing perfect alignment). This empirical evidence shows that instances where similar PageRank scores do not correspond to true structural symmetry are extremely rare in practice. Consequently, the impact on the overall model's performance is negligible, validating the effectiveness of our approximation approach.
>
> **Q2. The experimental tables (e.g., color-coded cells) may not fully convey information through text descriptions alone. It is recommended to add textual explanations in the paper, such as clarifying the meaning of "VI" values and specifying threshold ranges.**
>
> >R2: We appreciate the reviewer's suggestion. We will update the caption of Table 2 to the following:
>
> > _Table 2. Variation of Information (VI) between automorphic equivalence ($\simeq_{\text{auto}}$) and PageRank-based approximate equivalence ($\simeq_{\text{PR}}$) over eight benchmark datasets. Lower VI values (with 0 indicating perfect alignment) are shown in lighter colors._
>
> **Q3. The paper would benefit from a discussion on the potential limitations of the model.**
>
> >R3: We thank the reviewer for recommending a discussion on potential limitations. While our core contribution focuses on static homogeneous graphs, we acknowledge two related limitations. First, GALE currently assumes a static graph setting, meaning that for dynamic graphs with evolving structures or features, the equivalence classes may change over time, necessitating incremental updates to partitions—a challenge common to equivalence-based methods. Second, our present work targets homogeneous graphs; extending our equivalence definitions to heterogeneous graphs (e.g., those with multi-typed nodes/edges) would require additional design considerations, such as type-specific automorphism constraints. We consider these aspects as important directions for our future work.
>
> **Q4. Some typos: - Line 75: Change "noting" to "note". - Line 252: Change "indicates" to "indicating". - Line 308 (right column): Change "a" to "an".**
>
> >R4: We thank the reviewer for pointing out these typographical errors. We will correct these typos in the revised version of the paper.
>
> **Q5. The definition of approximate attribute equivalence does not appear to satisfy transitivity (i.e., u≃v and v≃w do not necessarily imply u≃w). Does this mean it does not form a strict equivalence relation? If so, it is recommended to use "similarity groups" instead.**
>
> >R5: We thank the reviewer for this valuable observation. We acknowledge that the current definition of approximate attribute equivalence does not satisfy transitivity—meaning it does not form a strict equivalence relation. In light of this, we agree that referring to these groups as "similarity groups" may be more appropriate and accurately reflects their nature. We will make the necessary modifications in subsequent revisions of the paper.

---

> > ### Comment · Reviewer_nMZa · 2025-04-08
> >
> > Thanks for the response. My main concerns have been addressed.  I keep my score of 4 Accept.

---

### Decision · Program_Chairs · 2025-05-01

**Decision:**

Accept (oral)

**Comment:**

This paper advocates for self supervised graph learning methods to encourages similar representations for automorphic nodes. The reviewers were very positive on the novelty and overall contribution of the method, hence I recommend acceptance.